# ABL001, a Bispecific Antibody Targeting VEGF and DLL4, with Chemotherapy, Synergistically Inhibits Tumor Progression in Xenograft Models

**DOI:** 10.3390/ijms22010241

**Published:** 2020-12-29

**Authors:** Dong-Hoon Yeom, Yo-Seob Lee, Ilhwan Ryu, Sunju Lee, Byungje Sung, Han-Byul Lee, Dongin Kim, Jin-Hyung Ahn, Eunsin Ha, Yong-Soo Choi, Sang Hoon Lee, Weon-Kyoo You

**Affiliations:** 1R&D Center, ABL Bio Inc., 2F, 16 Daewangpangyo-ro, 712 beon-gil, Bundang-gu, Seongnam-si, Gyeonggi-do 13488, Korea; donghoon.yeom@ablbio.com (D.-H.Y.); yoseob.lee@ablbio.com (Y.-S.L.); ilhwan.ryu@ablbio.com (I.R.); sunju.lee@ablbio.com (S.L.); byungje.sung@ablbio.com (B.S.); hanbyul.lee@ablbio.com (H.-B.L.); dongin.kim@ablbio.com (D.K.); jinhyung.ahn@ablbio.com (J.-H.A.); eunsin.ha@ablbio.com (E.H.); sang.lee@ablbio.com (S.H.L.); 2Department of Biotechnology, CHA University, Pangyo-ro 335, Bundang-gu, Seongnam-si, Gyeonggi-do 13488, Korea; yschoi@cha.ac.kr

**Keywords:** anti-angiogenesis, delta-like ligand, irinotecan, paclitaxel, therapeutic antibody, VEGF

## Abstract

Delta-like-ligand 4 (DLL4) is a promising target to augment the effects of VEGF inhibitors. A simultaneous blockade of VEGF/VEGFR and DLL4/Notch signaling pathways leads to more potent anti-cancer effects by synergistic anti-angiogenic mechanisms in xenograft models. A bispecific antibody targeting VEGF and DLL4 (ABL001/NOV1501/TR009) demonstrates more potent in vitro and in vivo biological activity compared to VEGF or DLL4 targeting monoclonal antibodies alone and is currently being evaluated in a phase 1 clinical study of heavy chemotherapy or targeted therapy pre-treated cancer patients (ClinicalTrials.gov Identifier: NCT03292783). However, the effects of a combination of ABL001 and chemotherapy on tumor vessels and tumors are not known. Hence, the effects of ABL001, with or without paclitaxel and irinotecan were evaluated in human gastric or colon cancer xenograft models. The combination treatment synergistically inhibited tumor progression compared to each monotherapy. More tumor vessel regression and apoptotic tumor cell induction were observed in tumors treated with the combination therapy, which might be due to tumor vessel normalization. Overall, these findings suggest that the combination therapy of ABL001 with paclitaxel or irinotecan would be a better clinical strategy for the treatment of cancer patients.

## 1. Introduction

Tumor angiogenesis, the formation of new blood vessels in solid tumors, plays an important role in tumor cell survival, growth, and metastasis [1]. A major driving force of tumor angiogenesis is the signaling pathway involving vascular endothelial growth factor (VEGF) and its receptors (VEGFRs) [2]. Several angiogenesis inhibitors, including antibodies and small molecule compounds targeting the VEGF/VEGFR signaling pathway, have been approved by the Food and Drug Administration (FDA), and used for the treatment of many different types of cancers [3]. Besides cancer treatment, VEGF/VEGFR inhibitors, including antibody fragments, aptamers, and VEGF-Traps were also approved and used for the treatment of ocular diseases caused by pathological angiogenesis [4,5,6,7,8]. VEGF/VEGFR blockade can inhibit VEGF-driven tumor angiogenesis, and the regression of tumor vessels is dependent on the VEGF signaling pathway. However, VEGF inhibitors alone are not capable of destroying all tumor blood vessels. In addition, preclinical studies indicate that VEGF inhibitors alone resulted in an increasingly aggressive and invasive pattern of tumors [9]. Some cancer patients are eventually refractory to anti-VEGF therapy, hence, next-generation angiogenesis inhibitors are being sought to augment the effects of VEGF inhibitors [10,11,12].

The DLL4/Notch signaling pathway can be a promising target of the next angiogenesis inhibitors, as this pathway regulates tumor angiogenesis with a different mechanism of action compared to that of the VEGF inhibitors [13,14,15]. Several preclinical xenograft studies have demonstrated that DLL4/Notch blockade inhibited tumor progression by promoting hyperproliferation of endothelial cells, which resulted in an increase in vascular density and a decrease in functional tumor vasculature [14,15,16,17,18,19,20]. DLL4/Notch inhibition is also known to reduce the number of cancer stem cells (CSCs), which are an important cancer cell population responsible for malignancy [21]. ABL001 is a bispecific antibody that simultaneously targets both DLL4 and VEGF, by linking each C-terminal of an anti-VEGF antibody (bevacizumab-similar) with a DLL4-binding single-chain Fv (scFv) [22,23]. In previous studies, ABL001 has demonstrated anti-cancer effects with higher potency in several human cancer xenograft models compared to that shown by the VEGF-targeting antibody (bevacizumab-similar) and the DLL4-targeting monoclonal antibody alone [23,24].

The safety and tolerability of ABL001 in cancer patients are now being evaluated in a phase 1 dose escalation study. The study was designed in a classical 3+3 dose-escalation schema where ABL001 is administered by IV across nine dose cohorts ranging from 0.3, 1, 2.5, 5, 7.5, 10, 12.5, 15, to 17.5 mg/kg biweekly [25]. No dose-limiting toxicity (DLT) was observed during the final cohort dose (17.5 mg/kg), and the maximum tolerated dose (MTD) was not reached. The most common treatment-related adverse events (AEs) (including all dose levels and all grades) were hypertension, anemia, anorexia, general weakness, and headache. However, they were well managed for all cohorts. Although the current phase 1 trial of monotherapy of ABL001 is ongoing, further clinical studies should be performed in combination with chemotherapy after the selection of optimal anti-cancer agents and cancer types. Since angiogenesis inhibitors target tumor endothelial cells, most VEGF/VEGFR blocking agents demonstrate clinical benefits for cancer patients when combined with chemotherapy [3]. Two different mechanisms of action of the combination therapy could provide synergistic anti-cancer efficacy for cancer patients. First, the combination therapy can destroy two separate components of tumors, tumor cells and tumor endothelial cells [26,27]. Second, the tumor vessel normalization by angiogenesis inhibitors enhances the delivery of cytotoxic anti-cancer agents [28,29]. However, the effects of a combination of ABL001 with chemotherapy on tumors and tumor blood vessels have not been fully studied. In this report, the in vivo anti-cancer effects of ABL001 with chemotherapy were evaluated in human gastric and colon cancer xenograft models and were compared to each monotherapy alone.

## 2. Results

### 2.1. Suppression of Tumor Progression in Various Cancer Xenograft Models by ABL001

To confirm the effects of ABL001 on tumor progression and to select the appropriate xenograft models for testing a combination treatment of ABL001 with chemotherapy, we evaluated the anti-cancer effects of ABL001 using several human gastric cancer (NUGC-3, MKN45, and SNU16 for mABL001, and GAPF006 for ABL001) xenograft models (Figure 1A), and human colon cancer (Colo205, WiDr, SW48, and SW620 for mABL001) xenograft models (Figure 1B). In the case of general xenograft models using human cancer cell lines, we used the mouse surrogate version of ABL001 (mABL001: binding to human VEGF and mouse DLL4) for the studies, as DLL4 is expressed by mouse endothelial cells involving tumor angiogenesis in tumor xenografts [23]. However, we used ABL001 in a patient-derived xenograft (PDX) model using GAPF006, which mimics the human tumor microenvironment from patients. Both bispecific antibodies, mABL001 and ABL001, inhibited tumor progression in the tested xenograft models at doses ranging from 1 to 6.5 mg/kg (Figure 1). The anti-cancer effects of mABL001 or ABL001 monotherapy were calculated as %TGI ranging from 27.4% to 57.2%, depending on the doses of mABL001 or ABL001 and cancer cell lines in xenograft models (Table 1). We focused on the dose level of ABL001 showing %TGI_50_ (50% tumor growth inhibition ratio) in each xenograft model because the dose of ABL001 and the xenograft model would be used for the combination therapy with paclitaxel or irinotecan. Based on the results from the dose range-finding studies, we selected GAPF006 gastric PDX, and SW48 or SW620 colon cancer xenograft models to address the efficacy of the combination treatment.

### 2.2. Synergistic Suppression on Tumor Progression by Combination Therapy

To determine whether the combination treatment of ABL001 with chemotherapy suppressed tumor progression with a higher strength as compared to that of each monotherapy, we evaluated the anti-cancer effects of the combination therapy using xenograft models compared to ABL001 or chemotherapy alone (Figure 2). In this study, we tested paclitaxel as chemotherapy in combination with ABL001 in gastric GAPF006 PDX (human gastric origin) xenograft, and irinotecan with mABL001 in SW48 or SW620 human colon tumor xenografts. In the gastric PDX model, the combination of paclitaxel and ABL001 demonstrated the most potent inhibition of tumor progression (74.75% TGI compared to 40.33% TGI in the paclitaxel-treated group and 46.20% TGI in the ABL001-treated group) (Figure 2A). Similarly, the combination of irinotecan with mABL001 suppressed tumor progression of SW48 and SW620 human colon cancer xenografts more potently compared to that by irinotecan or mABL001 alone (Figure 2B,C). At the endpoint of the SW48 xenograft study, the combination of irinotecan and mABL001 demonstrated 77.7% TGI, which was significantly different from the %TGI of the vehicle (*p* < 0.0001) group and irinotecan (*p* < 0.005) or mABL001 alone (*p* < 0.05) (Figure 2B). In the case of the SW620 xenograft model (human colon cancer), the combination treatment of irinotecan and mABL001 also exhibited the most potent anti-cancer effect (94.47% TGI) on tumor progression in the SW620 xenograft (Figure 2C).

### 2.3. More Potent Regression of Tumor Vessels by Combination Therapy

In order to evaluate the effects of the combination therapy on tumor blood vessels in xenograft models, the tumor vessels of SW620 tumor sections were analyzed using immunohistochemical staining for CD31 and VEGFR-2. Fluorescence microscopy images revealed that CD31-positive staining was localized in the vascular endothelial cells in the tumors (Figure 3A). The tumor vessel densities positive for CD31 in SW620 tumors treated with vehicle, irinotecan, mABL001, and combination were 0.71 ± 0.05%, 0.48 ± 0.03%, 0.36 ± 0.03%, and 0.18 ± 0.01%, respectively (Figure 3B). The percentage of positive area for CD31 in the combination was significantly lower than that of irinotecan or mABL001 alone. The area density of CD31-positive vessels in irinotecan-treated tumors was decreased by 32.4% and the density in mABL001-treated tumors was decreased by 49.3%, compared to the vehicle-treated group. However, the density of CD31-positive tumor vessels in the combination treatment decreased by 74.6% compared to the vehicle group (Figure 3B). VEGFR-2 was also strongly expressed on the endothelial cell membrane and cytoplasm in SW620 tumors (Figure 3A). The area densities of VEGFR-2-positive tumor vessels in the four groups were 0.65 ± 0.06%, 0.43 ± 0.04%, 0.23 ± 0.02%, and 0.13 ± 0.02%, respectively (Figure 3C). Compared to the vehicle-treated group, VEGFR-2-positive tumor vessels were reduced by 33.8% in the irinotecan-treated group, by 64.6% in the mABL001-treated group, and by 80% in the combination treatment group (Figure 3C). Based on the comparison of relative reduced levels between CD31-positive vessels with VEGFR-2-positive vessels in each tumor, VEGFR-2 expression was more reduced in tumor blood vessels compared to CD31 expression after VEGF blockade, mABL001 treatment, or the combination treatment (Figure 3B,C).

### 2.4. Decrease of DLL4 Expression on Tumor Vessels by Combination Therapy

DLL4 is expressed by tumor endothelial cells to regulate tumor angiogenesis, and by some tumor cells to maintain cancer stemness [14,15,21]. To address whether treatment with irinotecan, mABL001, or their combination affects DLL4 expression in tumors of xenograft models, DLL4 expression was examined using immunohistochemical staining using SW620 tumor sections from each group (Figure 4). DLL4 was mainly expressed on tumor blood vessels rather than tumor cells in this xenograft tumor and colocalized with CD31-positive tumor vessels (Figure 4A). The area densities of DLL4-positive tumor vessels were 0.40 ± 0.03% in vehicle, 0.24 ± 0.03% in irinotecan, 0.11 ± 0.02% in mABL001, and 0.05 ± 0.01% in the combination treatment group, respectively (Figure 4B). DLL4-positive tumor vessels were significantly reduced in the combination group compared to other groups. Compared to the vehicle group, DLL4-positive tumor vessels were reduced by 40% in the irinotecan group, by 72.5% in the mABL001 group, and by 87.5% in the combination group (Figure 4B). Similar to VEGFR-2 expression in tumor vessels, DLL4 expression was markedly reduced in tumor vessels compared to CD31 after treatment with mABL001 or the combination, rather than treatment with irinotecan alone (Figure 4B). Such a rapid reduction of DLL4 expression after mABL001 caused some tumor vessels to be stained only by CD31 but not by DLL4 (arrows in Figure 4A).

### 2.5. Increase of Tumor Apoptosis by Combination Therapy

Since the combination treatment of mABL001 with irinotecan showed more potent anti-cancer effects on tumor progression and anti-angiogenic effects on tumor vessels, the effects of the combination therapy on tumor cells were analyzed by immunohistochemical staining for activated caspase-3, an apoptotic cell marker. Immunofluorescence imaging revealed that activated caspase-3 was largely stained in the tumor cell nuclei rather than in the tumor endothelial cell nuclei in the tumor sections (Figure 5A). The area densities of activated caspase-3/DAPI positive cells were 5.16 ± 0.74% in the vehicle-treated group, 7.92 ± 1.05% in the irinotecan-treated group, 8.92 ± 1.65% in mABL001-treated group, and 10.87 ± 1.78% in the combination group (Figure 5B). The level of apoptotic tumor cells was significantly increased in the tumor sections after the combination treatment compared with the other groups. Such a potent increase in tumor cell apoptosis by the combination treatment might be due to direct cytotoxic effects of irinotecan against highly proliferating tumor cells together with the anti-angiogenic effects of mABL001, a bispecific antibody binding against dual antigens, VEGF, and mouse DLL4. The results suggest that the combination treatment of ABL001 with chemotherapy might provide better clinical benefits for cancer patients in clinical trials than ABL001 monotherapy.

## 3. Discussion

ABL001 (NOV1501/TR009), a bispecific antibody targeting VEGF and DLL4, is being developed as an anti-angiogenic cancer therapeutic that strengthens the effects of VEGF inhibitors and eventually overcomes resistance to anti-VEGF therapy [16,19,20,23]. ABL001 demonstrated more potent anti-angiogenic and anti-cancer effects in vitro and in vivo, as compared to the VEGF-targeting or the DLL4-targeting monoclonal antibodies alone, in various assay systems [23,24]. Based on the overall results of preclinical studies, the safety and tolerability of ABL001 are currently being tested with cancer patients previously treated heavily with chemotherapy or targeted therapy [25]. Other approved anti-angiogenic antibody therapeutics including bevacizumab, an antagonist of the VEGF ligand (VEGF-A: Avastin^®^), and ramucirumab, an antagonist of the VEGF receptor (VEGFR-2: Cyramza^®^), are generally used in a combination regimen with chemotherapy to treat cancer patients, providing more efficacious therapeutic options for cancer patients [3,30]. Anti-VEGF therapy is known to normalize tumor blood vessels, leading to a more efficient delivery of cytotoxic anti-cancer agents into tumor tissues [28,29], hence, most anti-VEGF therapy are used in the clinic in combination with chemotherapy [3,30]. Based upon the rationale mentioned above, newly developing VEGF/VEGFR inhibitors of monoclonal or bispecific antibodies, small molecule compounds, aptamers, and VEGF-Traps, have been evaluated synergistic anti-cancer effects with chemotherapy in various preclinical models before entering clinical trials [27,31,32].

Not only VEGF but DLL4 is also known to impair efficient delivery of anti-cancer drugs and to enhance chemoresistance in pancreatic cancer model due to induction of defective tumor angiogenesis [33]. However, little is known about the effects of a combination of ABL001, targeting dual antigens VEGF and DLL4, and chemotherapy on tumor vessels and tumor cells in xenograft models compared to each monotherapy alone. In this study, we evaluated the anti-angiogenic and anti-cancer effects of the combination treatment of ABL001 with paclitaxel or irinotecan in human gastric and colon cancer xenograft models.

The combination treatment of ABL001 with paclitaxel or irinotecan demonstrated more potent inhibition of tumor progression in these xenograft models, which is consistent with the previous report of the study collaborator [24]. Such potent anti-cancer effects of the combination therapy might be related to more significantly regressed tumor blood vessels, as compared to monotherapy with ABL001 or chemotherapy alone. Eventually, these anti-angiogenic and anti-cancer effects increased the apoptotic tumor status in the tumors post the combination treatment of ABL001 and chemotherapy. The underlying molecular mechanisms of action of the potent anti-cancer effects of ABL001 with chemotherapy might be due to the optimal combination effects of cytotoxic activity on tumor cells by paclitaxel or irinotecan together with more potent anti-angiogenic activity on tumor endothelial cells by ABL001, a VEGF, and DLL4 dual inhibitor. Moreover, because the VEGF-binding part of ABL001 is composed of the same IgG backbone and sequence as bevacizumab, ABL001 may have similar activity and function as bevacizumab in tumor vessels, resulting in a more effective delivery of anti-cancer agents, such as paclitaxel or irinotecan.

Based on the results of immunohistochemical analysis of tumor blood vessels, the expression levels of VEGFR-2 and DLL4, dual targets of ABL001, were markedly reduced in tumor endothelial cells after ABL001 treatment compared to that of CD31, a conventional endothelial cell marker. These findings are consistent with the previous results that VEGF blockade downregulates the levels of its receptor, VEGFR-2, and of DLL4 on endothelial cells [34,35]. Therefore, these results strongly support that the VEGF/VEGFR signaling pathway interacts with the DLL4/Notch signaling pathway in the tumor vasculature [35].

In addition to the cytotoxic anti-cancer agents, tumor vessel normalization by anti-VEGF therapy is also able to provide a better infiltration of immune cells, including cytotoxic T cells, into tumor tissues [36]. These reports suggest that anti-VEGF therapy can be the best option for combination therapy with immune checkpoint inhibitors for non-responsive cancer patients due to the lack of immune cells in the tumors, which are so-called ‘cold tumors’ or ‘non-inflamed tumors’. Indeed, a number of clinical studies for combination trials using anti-VEGF therapy with immune checkpoint inhibitors are ongoing for various cancer types [37]. During the past two years, 114 new combination regimens of VEGF and immune checkpoint inhibitors entered into clinical studies [38,39]. Among a large number of clinical studies, the FDA has approved several combination regimens of VEGF and immune checkpoint inhibitors, such as atezolizumab (an antagonist of PD-L1, Tecentriq^®^) plus bevacizumab with carboplatin and paclitaxel for the treatment of non-small cell lung cancer (NSCLC), avelumab (an antagonist of PD-L1, Bavencio^®^) plus axitinib (AG013736, a small molecule inhibitor of VEGFR tyrosine kinase, Inlyta^®^), and pembrolizumab (an antagonist of PD-1, Keytruda^®^) plus axitinib for the treatment of advanced renal carcinoma [40,41,42]. Recently, another combination regimen of atezolizumab plus bevacizumab was approved for the treatment hepatocellular carcinoma (HCC) as a first-line therapeutic option [43]. In this point of view, the results obtained in the current study imply that ABL001 may be another promising partner for combination therapy with immune checkpoint inhibitors, through the facilitation of immune cell infiltration via dual blockade of VEGF and DLL4 [28,29,33].

Currently, ABL001 is being tested for its safety, tolerability, and efficacy in phase 1 clinical studies with heavily pre-treated metastatic cancer patients. ABL001 has been well tolerated and no DLT is observed during dose escalation up to the final cohort, with manageable adverse effects generally exhibited by anti-cancer antibody therapeutics [25]. After the current dose escalation study of ABL001, further clinical development is scheduled to evaluate the efficacy of ABL001 in combination with chemotherapy. In conclusion, the results of this study provide important information for the clinical study design and plan for the combination treatment of ABL001 with chemotherapy.

## 4. Materials and Methods

### 4.1. Antibodies and Compounds

A human version of ABL001 bispecific antibody (ABL001) was produced under Good Manufacturing Practices (GMP) regulation by Bi-Nex (Incheon, Korea), and a mouse version of ABL001 bispecific antibody (mABL001) was produced by ABL Bio Inc., R&D Center (Gyeonggi-do, Seongnam-si, Korea), as described in a previous report [23]. Paclitaxel and irinotecan HCl were purchased from Hanmi Pharmaceutical Co. Ltd. (Seoul, Korea).

### 4.2. Cancer Cell Lines and Culture

Human gastric cancer cell lines, MKN45 (KCLB No.80103) and SNU16 (KCLB No.00016), were purchased from KCLB (Korea Cell Line Bank, Seoul, Korea), and NUGC-3 (JCRB0822) was obtained from JCRB (JCRB Cell Bank, Ibaraki, Japan). Human colon cancer cell lines, Colo205 (CCL-222), WiDr (CCL-218), and SW48 (CCL-231) were purchased from ATCC (American Type Culture Collection, Manassas, VA, USA). GAPF006 gastric cancer patient-derived tissues and SW620 human colon cancer cell line (LIDE, Shanghai, China) were also used for in vivo mouse xenograft studies. DMEM/F12, RPMI-1640, Leibovitz’s L-15, PBS, fetal bovine serum, 0.05% trypsin-EDTA, and antibiotic-antimycotic were purchased from Gibco (Carlsbad, CA, USA). Colo205, MKN45, SNU16, and NUGC-3 cells were cultured in RPMI-1640 culture medium containing 10% fetal bovine serum and antibiotic-antimycotic (1X). SW48 cells were cultured in DMEM/F12 culture medium containing 10% fetal bovine serum and antibiotic-antimycotic (1X). Colo205, MKN45, SNU16, NUGC-3, and SW48 cells were cultured in an incubator at 37 °C in a humidified atmosphere with 5% CO_2_ and 95% air. SW620 cells were cultured in Leibovitz’s L-15 medium containing 10% fetal bovine serum in an incubator at 37 °C in free gas exchange with atmospheric air.

### 4.3. Animals

Eight-week-old female BALB/c nu/nu mice (Orient Bio Inc., Gyeonggi-do, Korea) were used for the efficacy tests in Colo205, WiDr, MKN45, and SNU16 xenograft models, eight-week-old female CB17 SCID (Envigo, Indianapolis, IN, USA) were used for the efficacy tests in the SW48 xenograft model, and eight-week-old female BALB/c nu/nu mice (Beijing Vital River Laboratory Animal Technology Co., Ltd, Beijing, China) were used for the efficacy tests in the SW620 xenograft model and human gastric PDX (Patient-Derived Xenograft) model (LIDE). All animal experiments were approved by the Institutional Animal Care and Use Committee (IACUC), approval number: IACUC180067, approval date: 17 April 2018. Mice were maintained in a controlled environment (12 h light-dark cycle; temperature, 20–22 °C; 50–60% humidity), and ad libitum access to food and water.

### 4.4. Animal Studies

To evaluate the in vivo efficacy of mABL001, MKN45, SNU16, and NUGC-3 human gastric cancer cells (5 × 10^6^ cells/head) or Colo205, SW48, and WiDr human colon cancer cells (5 × 10^6^ cells/head) were implanted in the flank of BALB/c nu/nu mice or CB17 SCID mice. When the tumors had grown to an average volume of 150–200 mm^3^, the mice were divided into homogenous groups (6–12 mice/group), and treated with an intraperitoneal injection mABL001 (1.25, 2, 3.25, or 6.5 mg/kg), or ABL001 (GAPF006 PDX model, 6.5 mg/kg) twice per week. To evaluate the in vivo efficacy of mABL001 with chemotherapy, tumor growth was measured after treatment with the mouse version, mABL001 in SW48 or SW620 human colorectal cancer xenograft models, with or without irinotecan (20 or 40 mg/kg), respectively. BALB/c nu/nu mice were injected subcutaneously in the flank region with SW620 cells (5 × 10^6^ cells/head) in 0.1 mL of HBSS or GAPF006 tumor tissue fragments (9 mm^3^, approximately 50–90 mg), and CB17 SCID mice were injected subcutaneously in the flank region with SW48 cells (5 × 10^6^ cells/head). When the tumors had grown to an average volume of 150–200 mm^3^, the mice were divided into homogenous groups (7–10 mice/group). GAPF006 PDX model treated ABL001 (3.25 mg/kg) twice per week, and paclitaxel (15 mg/kg) was administered with an intraperitoneal injection once a week for three weeks. SW620 xenograft model treated mABL001 (2 mg/kg) twice per week, and irinotecan (40 mg/kg) were administered with an intraperitoneal injection once a week for three weeks.

Tumor size was measured twice per week using a caliper and then calculated using the formula, (length) × (width)^2^ × 0.5. When the average tumor size of the control group reached 2000 mm^3^, the treatment was stopped, and the mice were sacrificed to measure the tumor weight, and immunofluorescence analysis was performed (SW620 xenograft model). The efficacy was expressed as tumor growth inhibition [%TGI (mean volume of treated tumors/mean volume of control tumors) × 100]. Some mice were perfused with 4% paraformaldehyde in PBS for further immunofluorescence analysis of tumors.

### 4.5. Immunofluorescence Staining Analysis

To investigate whether mABL001 affects tumor angiogenesis and tumor cell survival, SW620 tumor sections were analyzed by immunofluorescence staining. For immunofluorescence staining analysis, SW620 tumors were removed from mice after cardiac perfusion and then embedded in OCT solution (Cat#3801480; Leica, Wetzlar, Germany) to produce frozen tumor blocks. The frozen tumors were sectioned (4-µm; Leica CM3050S; Leica) and permeabilized with washing buffer (PBS containing 0.03% Triton X-100) for 10 min, then blocked with 5% normal goat serum (Cat#S-1000; Vector Laboratories, Burlingame, CA, USA) or horse serum (Cat#16050122; Gibco) in the washing buffer. Tumor vessels were stained with rat anti-mouse CD31 (1:100, Cat#553370; BD, Franklin Lakes, NJ, USA) and goat anti-mouse VEGFR-2 antibody (1:100, Cat#AF644; R&D Systems, Minneapolis, MN, USA), respectively. Apoptotic cells in the tumors were stained with rabbit anti-mouse/human activated caspase-3 antibody (1:200, Cat#AF835; R&D Systems). DLL4 levels were detected with goat anti-mouse DLL4 antibody (1:100, Cat#AF1389; R&D Systems), which is cross-reactive (about 50%) with human DLL4. After being washed three times, the sections were stained for each secondary antibody, Alexa-568-conjugated goat anti-rat IgG (1:250, Cat#A11077), donkey anti-goat IgG (1:250, Cat#A11057), Alexa-488-conjugated goat anti-rabbit IgG (1:500, Cat#A11008), or donkey anti-rat (1:500 or 1:250, Cat#A21208), all from Thermo Fisher Scientific (Waltham, MA, USA). Stained tumors were mounted with Vectashield (Vector Laboratories) containing DAPI (4′,6-diamidino-2-phenylindole), and digital images of the tumors were captured using a Zeiss fluorescence microscope (Axio observer.7, Carl Zeiss, Oberkochen, Germany) with a camera (Axiocam, Carl Zeiss). Digital fluorescence images were analyzed using a Zeiss analysis software program (ZEN 2.6, Carl Zeiss).

### 4.6. Statistics

Graph creation and statistical analysis were performed using GraphPad Prism (GraphPad software Inc., San Diego, CA, USA) version 8.4.3. Values were expressed as the means ± SEM. Normality of data was tested using the Shapiro–Wilk test or Anderson–Darling test. Comparison between two groups was performed using the Student’s *t*-test. Multiple group comparisons were made parametric one-way ANOVA followed post hoc test (Tukey’s test, *p* < 0.0001, *p* < 0.001, *p* < 0.01, *p* < 0.05 values were considered as significant). The nonparametric Kruskal–Wallis test was used for the other cases.

## Figures and Tables

**Figure 1 ijms-22-00241-f001:**
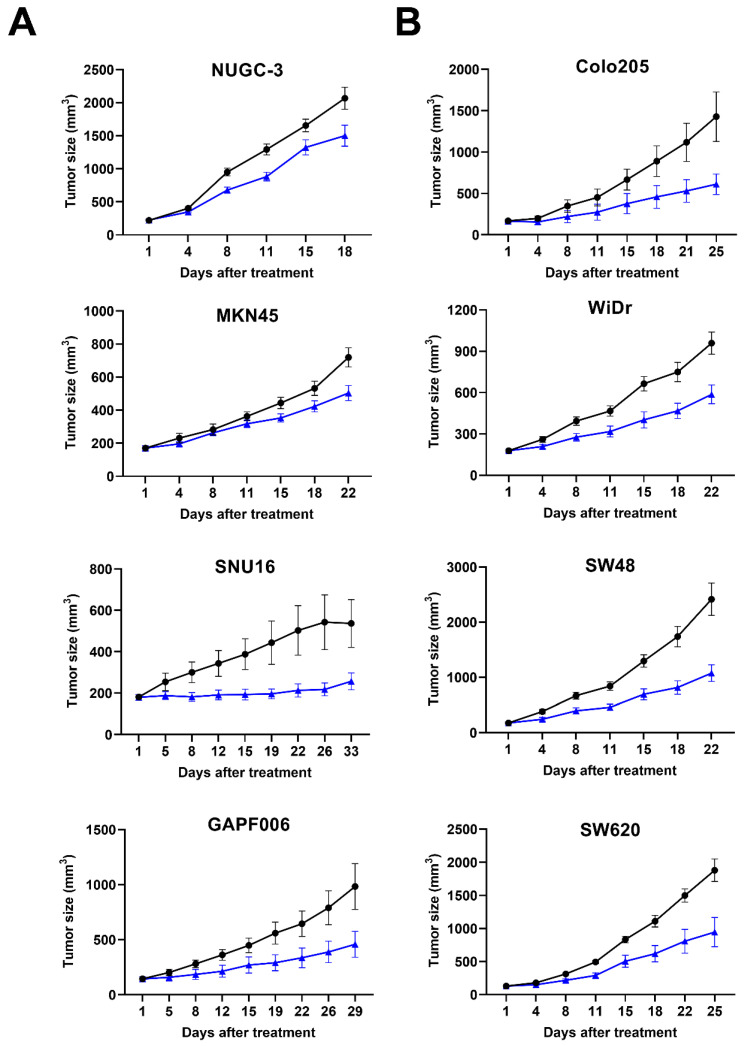
ABL001 strongly inhibited tumor progression of various human gastric and colon cancer xenograft models. Tumor size was measured twice per week and compared between vehicle (closed circle) and ABL001 (closed triangle) in human gastric cancer (NUGC-3, MKN45, SNU16 for mABL001, and human patient-derived gastric cancer GAPF006 for ABL001) xenograft model (**A**) and human colon cancer (Colo205, WiDr, SW48, SW620 for mABL001) xenograft model (**B**). ABL001 treatment significantly delayed tumor progression in different cancer xenograft models compared to control group of vehicle treatment. Error bars: mean ± SEM.

**Figure 2 ijms-22-00241-f002:**
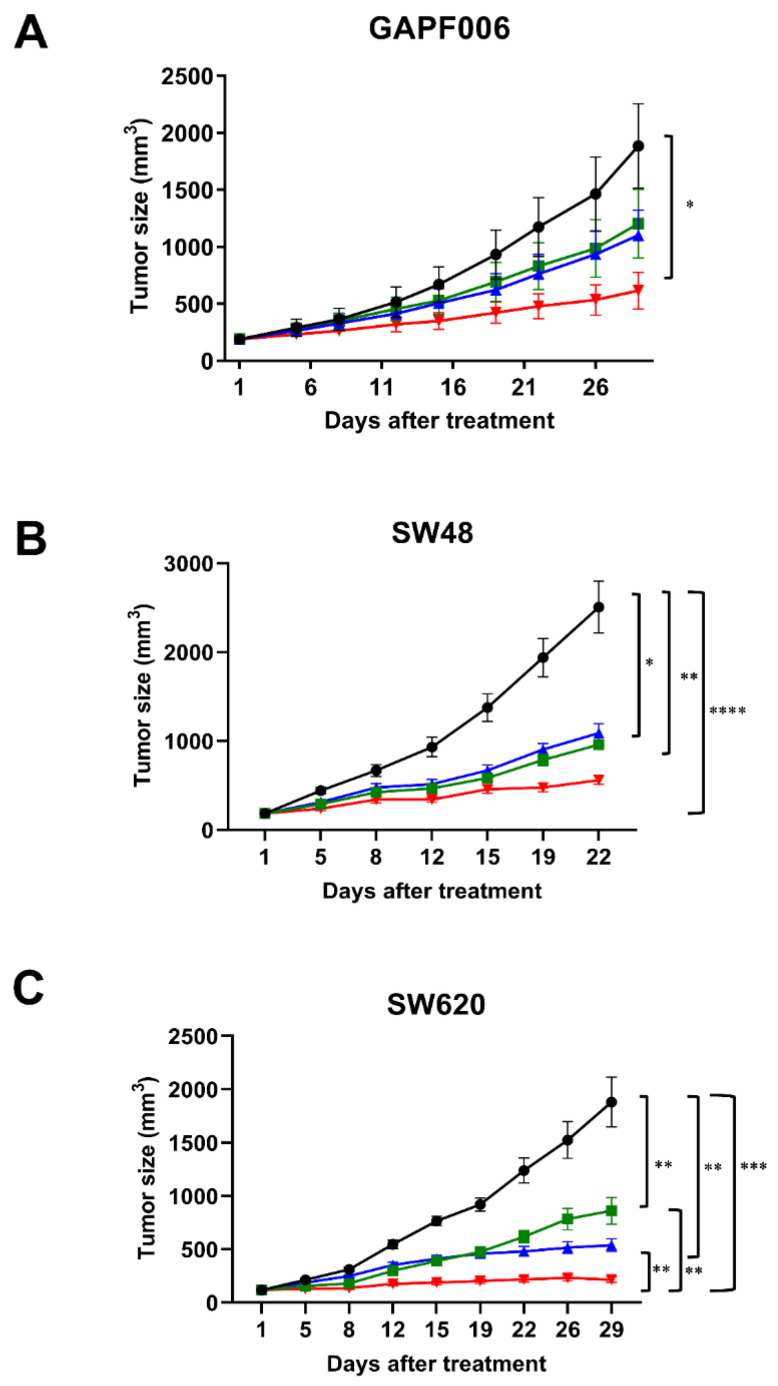
ABL001 in combination with chemotherapy with paclitaxel or irinotecan synergistically inhibited tumor progression in human gastric PDX and colon cancer xenograft models. In GAPF006 human gastric PDX model (**A**), mice were treated with vehicle (closed circle, black), paclitaxel alone (closed rectangle, green), ABL001 (closed triangle, blue), or a combination of ABL001 and paclitaxel (closed reverse triangle, red). Compared to vehicle, each treatment group inhibited tumor progression (40.33% TGI in paclitaxel, 46.20% TGI in ABL001, and 74.75% TGI in the combination treatment). In the studies using SW48 (**B**) and SW620 (**C**) colon cancer xenograft models, mice were treated with vehicle (closed circle, black), irinotecan alone (closed rectangle, green), mABL001 (closed triangle, blue), or a combination of mABL001 and irinotecan (closed reverse triangle, red). In the case of both colon cancer xenograft models, the combination treatment of mABL001 and irinotecan showed the most potent effects on tumor progression (77.7% TGI in SW48 and 94.47% TGI in SW620 xenograft models). Each line represents the average tumor size (mm^3^) of each treatment group ± SEM. * *p* < 0.05, ** *p* < 0.01, *** *p* < 0.001, **** *p* < 0.0001 by Tukey’s test.

**Figure 3 ijms-22-00241-f003:**
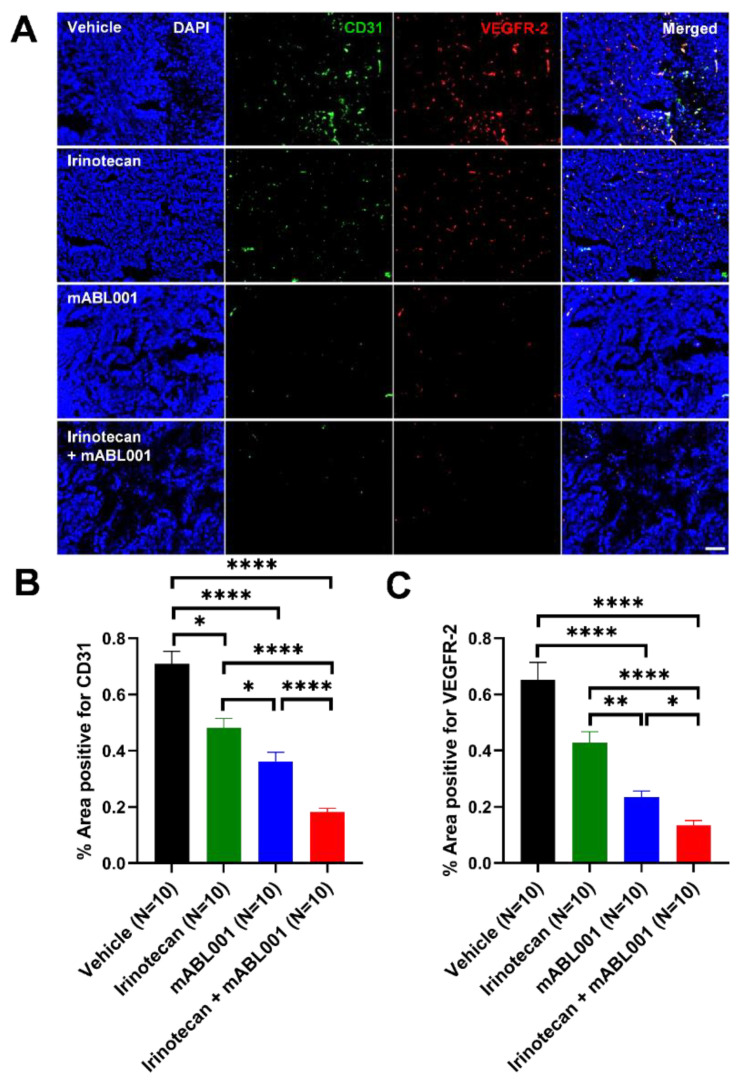
Combination therapy more potently regressed tumor blood vessels in SW620 xenograft model. Representative immunofluorescence images (**A**) show the tumor vasculature in SW620 tumor tissues stained for CD31, a generally conserved endothelial cell marker (green) and VEGFR-2 (red) with DAPI (blue). Most tumor blood vessels in vehicle group were stained and colocalized with both markers, CD31 and VEGFR-2. The area densities of CD31 (**B**) and VEGFR-2 (**C**) positive vessels were measured in each group. After irinotecan treatment, CD31 or VEGFR-2 positive tumor blood vessels were slightly regressed compared to vehicle treatment. However, after mABL001 or the combination treatment of mABL001 and irinotecan, CD31 and VEGFR-2 positive tumor vessels were significantly reduced (**B**,**C**). VEGFR-2 expression reduced more rapidly on tumor vessels. Scale bar indicates 200 µm. Error bars: mean ± SEM. * *p* < 0.05, ** *p* < 0.01, **** *p* < 0.0001 by Kruskal–Wallis test.

**Figure 4 ijms-22-00241-f004:**
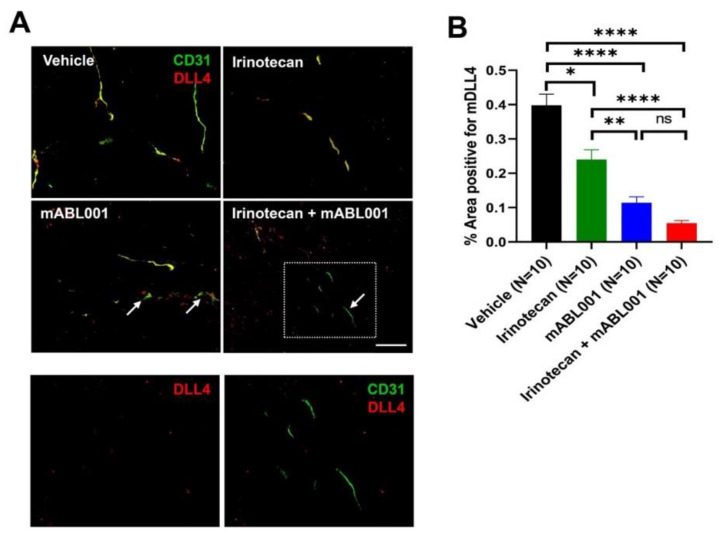
ABL001 significantly reduced DLL4 expression in tumor blood vessels. Representative immunofluorescence images (**A**) indicate the tumor vasculature in SW620 tumor tissues stained for CD31 (green) and DLL4 (red). The bottom figures (**A**) are magnified images of the dotted region of the combination treatment of mABL001 and irinotecan. The left image was shown only by red channel, whereas the right one was shown by merged channels (red and green). Similar to VEGFR-2, DLL4 was stained and colocalized on CD31 positive tumor blood vessels. The area density of DLL4 (**B**) positive vessels was measured in tumors of each group. Compared to vehicle or irinotecan treatment, DLL4 positive tumor vessels were significantly reduced in tumors after mABL001 or the combination treatment. Some tumor vessels were stained only for CD31 but not for DLL4, after mABL001 or the combination treatment group (arrows and dotted box in A). Scale bar indicates 50 µm in the bottom two images and 100 µm in the other images. Error bars: mean ± SEM. * *p* < 0.05, ** *p* < 0.01, **** *p* < 0.0001 by Kruskal–Wallis test.

**Figure 5 ijms-22-00241-f005:**
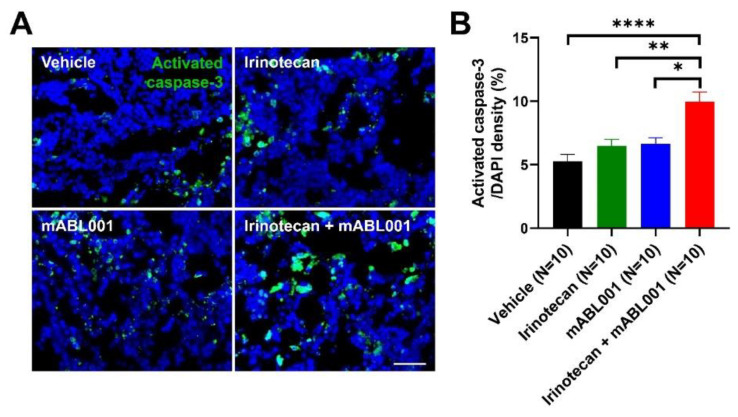
Combination therapy markedly increased apoptotic tumor cells in SW620 xenograft model. Representative immunofluorescence images (**A**) reveal apoptotic cells stained for activated caspase-3 (green) with DAPI (blue) in SW620 tumor tissues. The area densities of activated caspase-3-positive apoptotic cells were measured in each group (**B**). Apoptotic cells in tumors were marginally increased after irinotecan or mABL001 treatment, but the increase was not significant compared to vehicle treatment. However, the combination treatment of mABL001 and irinotecan markedly increased the apoptotic cell population in tumors. Scale bar indicates 50 µm. Error bars: mean ± SEM. * *p* < 0.05, ** *p* < 0.01, **** *p* < 0.0001 by Kruskal–Wallis test.

**Table 1 ijms-22-00241-t001:** Summarized information of animal studies using human gastric and colon cancer xenograft models.

Cancer Type	Cancer Cell Line	Dose (mg/kg)	Treatment Schedule	Animal Number (*n*/Group)	%TGI	*p* Value
Gastric	NUGC-3	1	Biweekly	11	27.4	0.0275
	MKN45	1.25		10	30.0	0.0378
	SNU16	3.25		12	52.2	0.0010
	GAPF006	6.5		10	53.3	0.0051
Colon	SW48	1.25	Biweekly	10	55.5	0.0264
	SW620	2		6	49.7	0.0224
	Colo205	3.25		8	57.2	0.0177
	WiDr	6.5		9	38.8	0.0131

GAPF006, gastric patient-derived xenograft model, %TGI = tumor growth inhibition, *p* value: Student’s *t*-test.

## Data Availability

Data is contained within the article.

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
