# Peer review of "ABL001, a Bispecific Antibody Targeting VEGF and DLL4, with Chemotherapy, Synergistically Inhibits Tumor Progression in Xenograft Models"

_ijms, 2020, doi:10.3390/ijms22010241_

Round 1

Reviewer 1 Report

The study is very well organized and scientifically sound. I've few minor comments. Post the addressal of the comments, the manuscript can be accepted.

1)Line 36: The author must add the following references and mention about aptamers that are also used for VEGF inhibition.
1) Harleen Kaur, John G Bruno, Amit Kumar, Tarun Kumar Sharma; Aptamers in the Therapeutics and Diagnostics Pipelines. Theranostics. 2018.

2) Ruckman J, Green LS, Beeson J, Waugh S, Gillette WL, Henninger DD, et al. 2'-Fluoropyrimidine RNA-based aptamers to the 165-amino acid form of vascular endothelial growth factor (VEGF165). Inhibition of receptor binding and VEGF-induced vascular permeability through interactions requiring the exon 7-encoded domain. J Biol Chem. 1998; 273: 20556-67.

2) The authors must carry an experiment comparing bispecific antibody, monocloanl antibody, small molecule, aptamer and different combinations of the aforementioned molecules against VEGF to make the results more convincing.

Author Response

Title: ABL001, a bispecific antibody targeting VEGF and DLL4, with chemotherapy, synergistically inhibits tumor progression in xenograft models [IJMS-1048436]

Overall, all comments and suggestions from reviewers are insightful, which make the manuscript more solid. On behalf of all authors, I thank both reviewers for their valuable contributions. We did our best to address all issues raised from reviewers as below. All changes and revision were highlighted as yellow background in the revised manuscript for reviewers’ easy and clear second reviewing process.

The study is very well organized and scientifically sound. I've few minor comments. Post the addressal of the comments, the manuscript can be accepted.

1)Line 36: The author must add the following references and mention about aptamers that are also used for VEGF inhibition.

1) Harleen Kaur, John G Bruno, Amit Kumar, Tarun Kumar Sharma; Aptamers in the Therapeutics and Diagnostics Pipelines. Theranostics. 2018.

2) Ruckman J, Green LS, Beeson J, Waugh S, Gillette WL, Henninger DD, et al. 2'-Fluoropyrimidine RNA-based aptamers to the 165-amino acid form of vascular endothelial growth factor (VEGF165). Inhibition of receptor binding and VEGF-induced vascular permeability through interactions requiring the exon 7-encoded domain. J Biol Chem. 1998; 273: 20556-67.

Response: We mentioned that aptamers have been approved and used for ocular disease therapy as another class of VEGF inhibitors together with VEGF-targeting antibody fragments and VEGF-Traps. We also included the references suggested by reviewer #1 with additional new references about VEGF-targeting antibody fragments and VEGF-Traps (line 37-39 in introduction).

2) The authors must carry an experiment comparing bispecific antibody, monocloanl antibody, small molecule, aptamer and different combinations of the aforementioned molecules against VEGF to make the results more convincing.

Response: We agreed with reviewer #1’s comments that more preclinical studies using different classes of VEGF inhibitors and their combinations will make the results of the manuscript more convincing. However, the editor of the journal office gave us only for 5 days to submit the revised manuscript. Since we think it is too short time to perform actual animal experiments, we added additional sentence describing that VEGF inhibitors of many different modalities have been testing with chemotherapy in various preclinical models (line 236-239 in discussion). We also added the related references.

Reviewer 2 Report

In the manuscript entitled “ABL001, a bispecific antibody targeting VEGF and DLL4, with chemotherapy, synergistically inhibits tumor progression in xenograft models” Dong-Hoon Yeom and co-workers have done synergistic studies with ABL001 (a bispecific antibody that simultaneously targets both DLL4 and VEGF) and approved anti-cancer drugs such Paclitaxel and Irinotecan in human gastric or colon cancer xenograft models. The synergistic effects were found to be better than each monotherapy. The combination of ABL001 and Irinotecan also markedly increased apoptotic tumor cells in SW620 xenograft model. As ABL001 is currently in phase 1 of clinical studies so these studies could be useful in finding better combination therapy for the treatment of gastric or colon cancer. Therefore, my decision goes in favor of publication of this article in IJMS after minor revision.

Queries:

  • The introduction needs to be written in more detail by giving more emphasis on combination therapy.

Author Response

In the manuscript entitled “ABL001, a bispecific antibody targeting VEGF and DLL4, with chemotherapy, synergistically inhibits tumor progression in xenograft models” Dong-Hoon Yeom and co-workers have done synergistic studies with ABL001 (a bispecific antibody that simultaneously targets both DLL4 and VEGF) and approved anti-cancer drugs such Paclitaxel and Irinotecan in human gastric or colon cancer xenograft models. The synergistic effects were found to be better than each monotherapy. The combination of ABL001 and Irinotecan also markedly increased apoptotic tumor cells in SW620 xenograft model. As ABL001 is currently in phase 1 of clinical studies so these studies could be useful in finding better combination therapy for the treatment of gastric or colon cancer. Therefore, my decision goes in favor of publication of this article in IJMS after minor revision.

Queries:

The introduction needs to be written in more detail by giving more emphasis on combination therapy.

Response: We agreed with reviewer #2’s comments and added additional sentences giving scientific rationale and background on the combination therapy between VEGF inhibitors and chemotherapy (line 66-72 in introduction). We also added the related references.
